# Many-body enhancement of high-harmonic generation in monolayer MoS$_2$

Victor Chang Lee [1,2], Lun Yue [3], Mette B. Gaarde[3], Yang-hao Chan [4,5] & Diana Y. Qiu [1,2]

Many-body effects play an important role in enhancing and modifying optical absorption and other excited-state properties of solids in the perturbative regime, but their role in high harmonic generation (HHG) and other nonlinear response beyond the perturbative regime is not well-understood. We develop here an ab initio many-body method to study nonperturbative HHG based on the real-time propagation of the non-equilibrium Green's function with the GW self energy. We calculate the HHG of monolayer MoS$_2$ and obtain good agreement with experiment, including the reproduction of characteristic patterns of monotonic and nonmonotonic harmonic yield in the parallel and perpendicular responses, respectively. Here, we show that many-body effects are especially important to accurately reproduce the spectral features in the perpendicular response, which reflect a complex interplay of electron-hole interactions (or exciton effects) in tandem with the many-body renormalization and Berry curvature of the independent quasiparticle bandstructure.

High harmonic generation (HHG) is a nonlinear optical process where a system illuminated by an intense laser pulse emits high-energy photons at frequencies that are harmonics of the driving field[1]. HHG forms the basis of attosecond science through the generation of coherent, attosecond light pulses[2,3] and the concomitant study and control of ultrafast electronic and atomic processes with unprecedented temporal and spatial resolution[4]. While historically HHG has focused on the gas phase, in recent years, there has been rapid progress in the observation and control of HHG in solids. In 2011, harmonics up to the 25th order were generated in ZnO from a mid-infrared (mid-IR) pulse[5], and since then, HHG has been observed across a wide variety of solid materials[4,6], including wide bandgap dielectrics[7–9], bulk semiconductors[10–13], noble gas solids[14], strongly correlated materials[15], and low-dimensional and nanostructured materials[16–22] where many-body effects are enhanced due to a combination of confinement and reduced screening.

In the gas phase, HHG is widely understood within a semiclassical three-step model: an electron tunnels into and is accelerated in the free-electron continuum, eventually emitting a high-energy photon when it recombines with a bound hole[23–27]. In solids, the excited electron is not a free electron but instead explores the conduction band manifold, resulting in additional sources of nonlinear current due to Bloch oscillations and anomalous velocities associated with the Berry curvature of the bands[5,6,10,27]. Thus, in solids, HHG holds promise not just as a light source but also as a dynamical, all-optical, and attosecond-resolution probe of the band structure with the potential to target specific bands under ambient pressures and in operando conditions. However, the theoretical interpretation of such spectra remains challenging, and the role of many-electron correlations and exciton effects, which dominate both linear and nonlinear spectroscopies in the low-field regime, are all but unknown.

In experiment, there are indications that exciton effects beyond the independent-particle (IP) picture may play an important role in the high-harmonic spectra of solids. For instance, the second plateau seen in HHG of noble gas solids corresponds to twice the energy of the lowest-energy optically-bright exciton[14], and odd harmonics in monolayer MoS$_2$ are enhanced compared to the bulk[16], hinting at parallels to linear absorption where the oscillator strength is similarly enhanced due to the larger exciton binding energy in the monolayer limit[28–32]. Nonetheless, the preponderance of existing theory assumes an IP picture, with the justification that a bound exciton will dissociate

[1]Department of Mechanical Engineering and Materials Science, Yale University, New Haven, CT, USA. [2]Energy Science Institute, Yale University, New Haven, CT, USA. [3]Department of Physics and Astronomy, Louisiana State University, Baton Rouge, LA, USA. [4]Institute of Atomic and Molecular Sciences, Academia Sinica, Taipei, Taiwan. [5]Physics Division, National Center of Theoretical Sciences, Taipei, Taiwan. ✉e-mail: yanghao@gate.sinica.edu.tw; diana.qiu@yale.edu

under the strong laser field required for HHG–the so-called strong-field approximation (SFA)[26,33,34]. An IP picture already captures many spectral features of HHG in solids, including selection rules for polarization and crystal orientation[10,13,16], scaling of the energy cutoff[5], multiband effects[10,21], and Berry curvature[16,34]. However, incorporation of a model Coulomb interaction that reproduces the exciton binding energy in $SiO_2$ within the semiconductor Bloch equation (SBE) formalism suggests that excitons can significantly change the relative contributions of the interband and intraband current[35]. Excitons have also been shown to qualitatively change HHG in Mott insulators in a Hubbard model with a driving field[36], and in both transition metal dichalcogenides (TMDs) and a model one-dimensional solid using a time-dependent Hartree-Fock approach[37–39]. Despite these indications of the importance of exciton effects, a nonperturbative, fully ab initio theory that can rigorously describe many-electron interaction effects in the higher-order nonlinear response of real materials is still missing.

Here, we develop such a theory based on the Keldysh formalism for nonequilibrium Green's functions, which we refer to as time-dependent adiabatic GW (TD-aGW), and apply it to understand HHG in monolayer $MoS_2$, a prototypical two-dimensional (2D) semiconductor, where exciton effects in the linear[29,32] and perturbative nonlinear[40–44] regimes are known to be large. Remarkably, we find that many-body interactions are key to capturing the relative strength of harmonics in the direction perpendicular to the driving field, while harmonics parallel to the driving field remain qualitatively similar to the non-interacting IP picture. The importance of many-body effects in the perpendicular configuration is a consequence of the interplay of an excitonic enhancement of the oscillator strength for interband transitions in tandem with the intraband anomalous velocity arising from the Berry curvature, which can drive the electron and hole in the same direction in real space, leading to the potential for enhanced electron-hole correlation effects. Our results are in good agreement with experimental spectra[16,18,21], which show significant nonmonotonic behavior of the harmonic yield in the perpendicular direction, an effect that we reveal cannot be reproduced at the correct harmonics in a noninteracting picture.

## Results and discussion

We start by performing a GW plus Bethe Salpeter equation (GW-BSE)[45–47] calculation of the equilibrium quasiparticle and optical properties of monolayer $MoS_2$ within ab initio many-body perturbation theory (MBPT) to obtain the equilibrium starting point for our TD-aGW calculation. We find the direct bandgap is 1.67 eV at the density functional theory (DFT) level, within the generalized gradient approximation of Perdew, Burke and Ernzerhof (PBE)[48], and increases to 2.54 eV after including the electron correlations at the one-shot $G_0W_0$ level in good agreement with previous calculations[29,49]. Figure 1a, b shows the linear absorption spectrum at three different levels of theory: the GW-BSE approach, the DFT level (IP@DFT-PBE), and the GW level within the random phase approximation (IP@GW-RPA). The linear absorption at the TD-aGW level is identical to the GW-BSE level[44]. The exciton binding energy of the lowest optically active exciton (peak A) is 0.5 eV, which agrees well with previous calculations[29,49]. Additional computational details can be found in Supplementary Note 2.

Starting from the equilibrium $G_0W_0$ energies, we then simulate the time-dependent current and HHG spectrum of monolayer $MoS_2$ within TD-aGW for a laser field with a peak intensity of 1.5 $TW/cm^2$, a wavelength of 3.7 $\mu$ m (0.335 eV), and a pulse duration of 100 fs, which corresponds to the experimental conditions in ref. 21. At this wavelength, the 6th harmonic is resonant with the A exciton peak in monolayer $MoS_2$. We rotate the polarization of the driving field over different fundamental angles ($\theta$) with respect to the $MoS_2$ mirror plane and then calculate the nonlinear current and corresponding HHG spectra in the direction parallel to the driving field (parallel configuration) or perpendicular to the driving field (perpendicular configuration). We compare the results from TD-aGW with an IP picture obtained from the time

propagation of the density matrix without the time-dependent electron self-energy term ($\delta\Sigma(t)$ in Eq. (2)) and consider both cases where the equilibrium energies are treated at the $G_0W_0$ level (IP@GW) and the DFT level (IP@DFT) to understand the role of bandgap renormalization on HHG independent of electron-hole correlation effects.

The selection rules for the presence or absence of certain harmonics at certain angles is consistent across all levels of theory (Fig. 2). In the parallel configuration, all harmonics are allowed at a fundamental angle of $\theta = 0°$, but only odd harmonics are allowed at $\theta = 30°$. In the perpendicular configuration, which contains contributions from the anomalous current arising from the Berry curvature[34], all harmonics are forbidden at 0°, and only even harmonics are allowed at 30°. These selection rules can already be understood from a semi-classical analysis of the electronic motion in a single band in conjunction with the three-fold crystal symmetry, as explained by Liu et al. in ref. 16, and are not changed by many-body effects.

The inclusion of electron-hole interactions within TD-aGW results in an enhancement of certain harmonics in the HHG spectra. For visual clarity, we extract the HHG spectrum at 30°, in the parallel configuration and in the perpendicular configuration (Fig. 3). In the parallel configuration, we see a nearly monotonic decrease for the odd series of harmonics at all levels of theory (Fig. 3a, b). There are small enhancements in the harmonic intensity at energies that correspond to peaks in the linear absorption (e.g., the slight plateau around the 9th and 11th harmonics at the IP@DFT level, just above peak C), but the qualitative features of the spectra are similar at all levels of theory, suggesting that the primary role of many-body effects is in renormalizing the transition energies and changing the available density of states and the interband polarization due to the presence of bound excitons, leading to quantitative but not qualitative changes in the HHG spectrum.

For the even harmonics, on the other hand, experimental HHG spectra of monolayer $MoS_2$ show significant deviations from monotonic behavior[16,18,21,50]. Intriguingly, in our calculated spectra, only the TD-aGW calculation reproduces this strongly nonmonotonic behavior of the even harmonics in the perpendicular configuration under the current excitation conditions, suggesting that it may stem from many-body effects and cannot be understood from resonant enhancement alone. Within TD-aGW, with a 0.335 eV driving field at 30°, the 10th harmonic, which is above the energy of exciton peak C, is significantly enhanced compared to the 8th harmonic. However, the two noninteracting theories, despite still having large peaks in the linear absorption at peak C (Fig. 1a), both show a monotonic decrease in harmonic intensity with harmonic order, though the 10th harmonic is also slightly enhanced at the DFT level. We note that some spectral features may be sensitive to the dephasing time. When the dephasing time is decreased additional nonmonotonic features appear in both the parallel and perpendicular configurations, but the enhancement of the 10th harmonic at the TD-aGW level is robust to changes in the dephasing.

In Fig. 4, we show our calculated results overlaid with data points extracted from experiment. In the experiment performed by Liu et al.[16] in the perpendicular configuration, the 10th harmonic is much fainter than the 8th and 12th harmonics, while in the parallel configuration, there is a monotonic decrease in harmonic yield. In the experiment perform by Yoshikawa et al.[18], the 12th harmonic is enhanced compared to the 8th, 10th and 14th harmonics, while the odd harmonics decrease monotonically. We see that TD-aGW gives good agreement with experiment under both these excitation conditions, while the two IP levels of theory are not able to reproduce the experimental behavior. Comparing to the experiment done by Liu et al.[16], the IP@GW calculation pushes the nonmonotonic enhancement to the 14th harmonic in the perpendicular configuration, and the IP@DFT calculation shows an incorrect enhancement of the 11th and 13th harmonics in the parallel configuration. Furthermore, we note that the intensity of the 13th harmonic in the parallel configuration is too faint to resolve in experiment. Theoretically, only the TD-aGW predicts a sharp decay of the 13th harmonic, which is also observed in this

experiment. Comparing to the experiment done by Yoshikawa et al., the TD-aGW calculation also shows better agreement with the experiment, reproducing the enhancement of the 12th harmonic compared to the 10th and 14th harmonics. We note that the experimental data was originally reported in arbitrary units and has been scaled to match the highest yield harmonic in our calculations.

We can gain some understanding of the different behavior of the parallel and perpendicular harmonics by analyzing the different mechanisms driving the motion of the electrons and the holes in the direction parallel and perpendicular to the field at 30°[50–52].

In the parallel configuration, electrons and holes are driven in opposite directions by the laser field, and thus, the importance of electron-hole interactions is greatly diminished. In contrast, perpendicular to the driving field, the Berry curvature acts as a pseudo-magnetic field driving an anomalous intraband current for the electrons and holes[16,53]. Therefore, the electron and hole can move in the same direction in real space depending on their respective Berry curvature, and there is an exciton drift perpendicular to the electric field whenever the *exciton's* Berry curvature is non-zero[54]. Thus, the electron-hole interaction cannot be neglected even in the strong field regime, and as we see in our TD-aGW calculation, inclusion of such many-body effects are necessary to capture qualitative features of the HHG spectra.

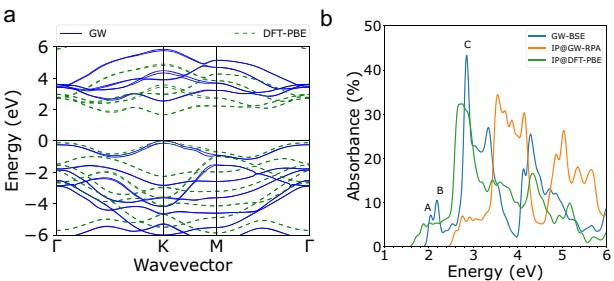

**Fig. 1 | Band structure and linear absorption spectra of MoS₂. a** Bandstructure of monolayer $MoS_2$ computed at the DFT (dashed green lines) and GW (solid blue lines) levels. **b** Linear absorption spectra computed with electron-hole interactions at the GW-BSE level (blue), at the noninteracting GW-RPA level (IP@GW-RPA, orange), and the DFT level (IP@DFT-PBE, green). Exciton peaks computed within GW-BSE are labelled A, B and C.

Additionally, we see that the nonmonotonic enhancement of specific harmonics does not correspond perfectly to the resonant energies of the exciton peaks. To understand this behavior, we look at the electron population in reciprocal space at different times over a laser field cycle from our TD-aGW simulation as shown in Fig. 5a–b for the lowest conduction band (CB) and Fig. 5c, d for the highest valence band (VB). Similar data for the IP picture is shown in the Supplementary Fig. 5. The total excited-state population computed within TD-aGW is greater than that in the IP@GW and IP@DFT approximations, which suggest that the inclusion of exciton effects has an important role in assisting the creation of carriers, consistent with previous Hartree Fock calculations[39].

The electron population is mostly localized near the K and K' valleys (with a triangular shape due to trigonal warping) over the entire cycle. The holes, on the other hand, also have a considerable population near Γ. The distribution of the electrons and holes in reciprocal space correlates with the distribution of the Berry curvature of the conduction and valence bands (Fig. 5e, f). Electrons(holes) initially excited at the K valley will have a negative(positive) Berry curvature that drives the motion perpendicular to the laser field. However, when the Berry curvature changes sign, the electrons or holes will change direction leading to a complex oscillatory motion in real space.

The electrons experience a rapid change in sign in the Berry curvature going from the K or K' valley to the secondary valleys along the path from K to Γ to K' (sometimes called the Q and Q' valleys respectively). The holes on the other hand, see a Berry curvature of the same sign everywhere on the path from K or K' to Γ. In reciprocal space, the coupling of intraband motion with interband recombination (driven by the real-space motion) confines the electrons near the K and K' valleys, while allowing the holes to populate both the K and Γ valleys (Fig. 5a–d). The electron localization in reciprocal space prevents the electron from entering the band nesting region, in the Q valley, where both the density of states and the linear absorption are maximized. Consequently, the HHG enhancement actually happens at energies higher than the maximum in the linear absorption (peak C), since they never reach the energy minimum in the Q valley. This effect is even more pronounced in the IP picture (see Supplementary Fig. 5), as the electron-hole interaction serves to localize the electron and hole in real space, resulting in a slightly increased delocalization in k-space.

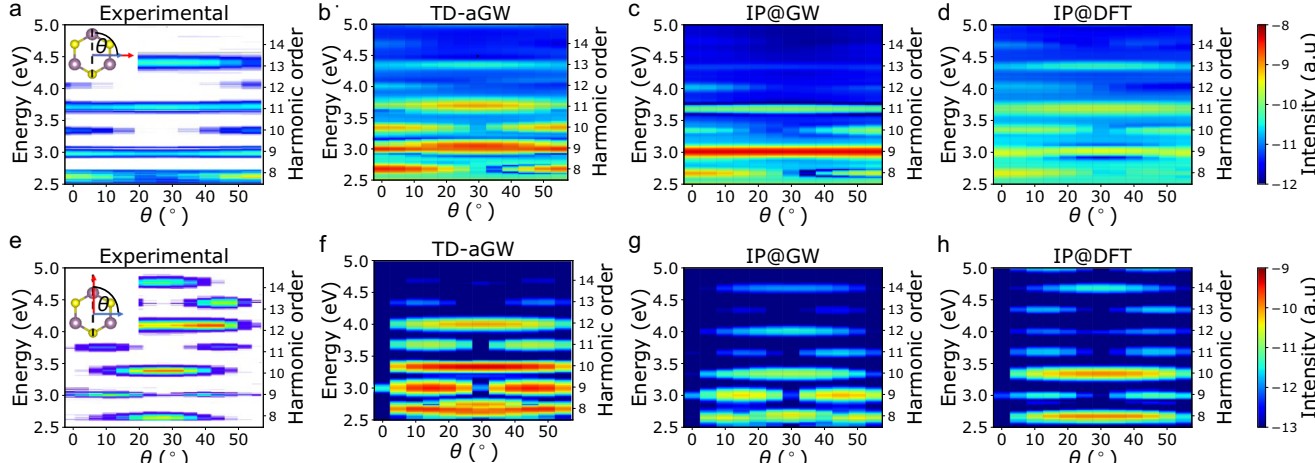

**Fig. 2 | Experimental and theoretical HHG anisotropy spectra. a** Experimental HHG reproduced with permission from ref. 21 in the parallel configuration. Inset shows the polarization of the driving field (red arrow) at an angle θ with respect to the mirror plane (black dashed line) and the direction of the measured HHG (blue arrow). **b** Angle-resolved HHG computed within TD-aGW, **c** at the independent-particle level with GW energies (IP@GW), and **d**) at the independent-particle level with DFT energies (IP@DFT). **e–h** Same as **a–d** in the perpendicular configuration. The calculated HHG intensity is presented on a logarithmic scale in atomic units (a.u.). Experimental data taken from ref. 21 is presented in arbitrary units that have been rescaled within each harmonic.

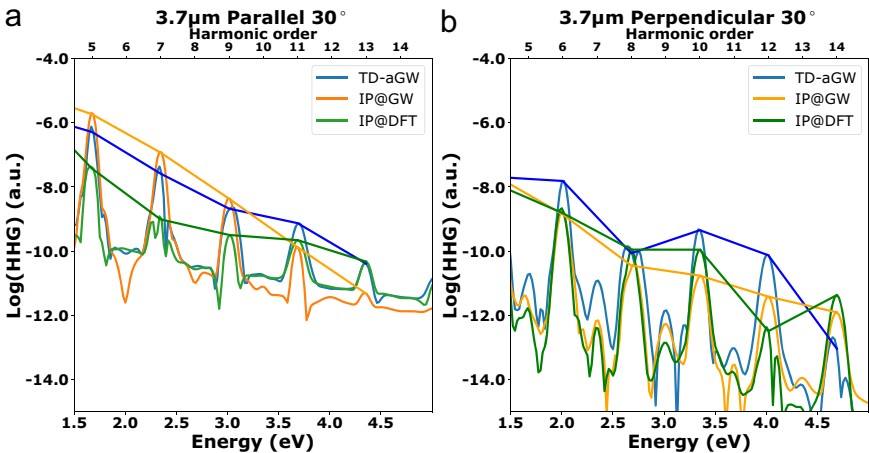

**Fig. 3 | Comparison of simulated HHG spectra at 30°.** The data is plotted for TD-aGW in blue, IP@GW in orange and IP@DFT in green at 30° for the **a** parallel configuration and **b** perpendicular configuration. The HHG intensity is presented in a log scale in atomic units (a.u.). A line connecting odd harmonic peaks in the parallel configuration and even harmonic peaks in the perpendicular configuration is provided as a guide to the eye.

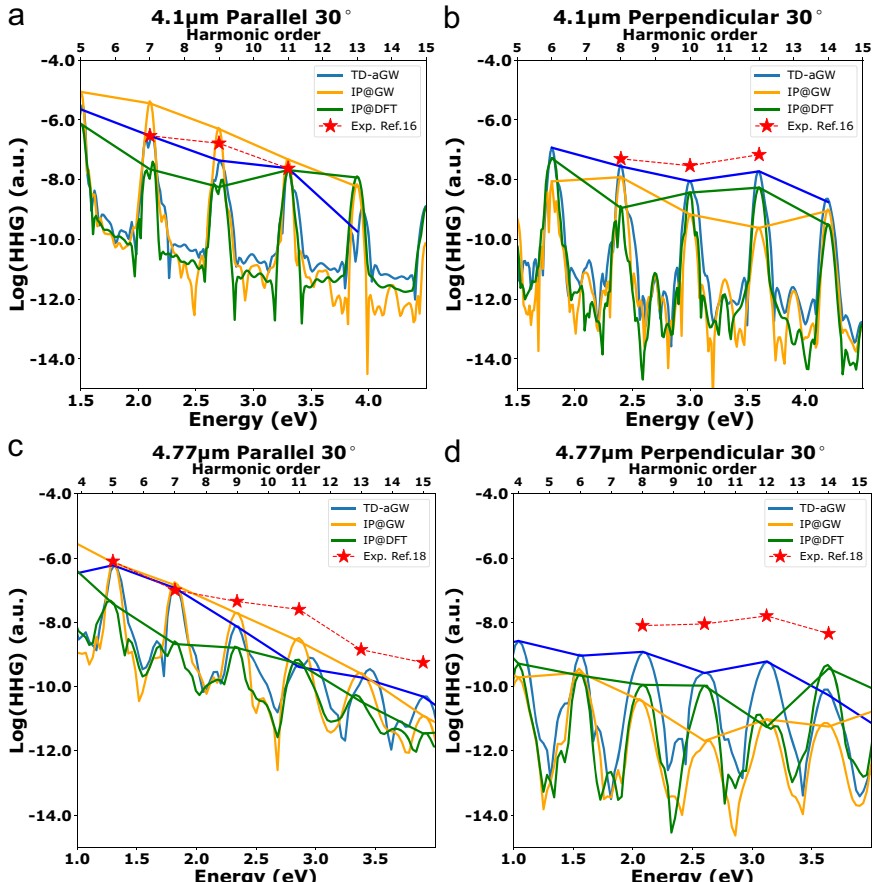

**Fig. 4 | Comparison of theoretical HHG spectra with previous experimental measurements.** The data shown is simulated with the same field as the experiment of Liu et al.[16] for the harmonics at 30° in the **a** parallel configuration and **b** perpendicular configuration. **c**, **d** show the comparison of the high harmonic spectrum simulated with the same field as Yoshikawa et al.[18] for the harmonics at 30° in the **c** parallel configuration and **d** perpendicular configuration. TD-aGW results are in blue, IP@GW in orange, and IP@DFT in green. Extracted experimental data is shown as red stars and scaled to match the amplitude of the highest intensity harmonic. The calculated HHG intensity is presented in a log scale in atomic units (a.u.).

The Berry curvature projected on the full band structure can be found in Supplementary Fig. 3.

In summary, we have developed an ab initio many-body method based on the real-time propagation of non-equilibrium Green's functions for the study of HHG. This method includes both non-local screening and electron-hole interactions at the level of GW-BSE, extending the state-of-the-art for linear response spectra to the non-perturbative regime. We find good agreement between our calculated spectra and experimental spectra of monolayer MoS₂, a prototypical 2D semiconductor known to host strong many-body exciton effects.

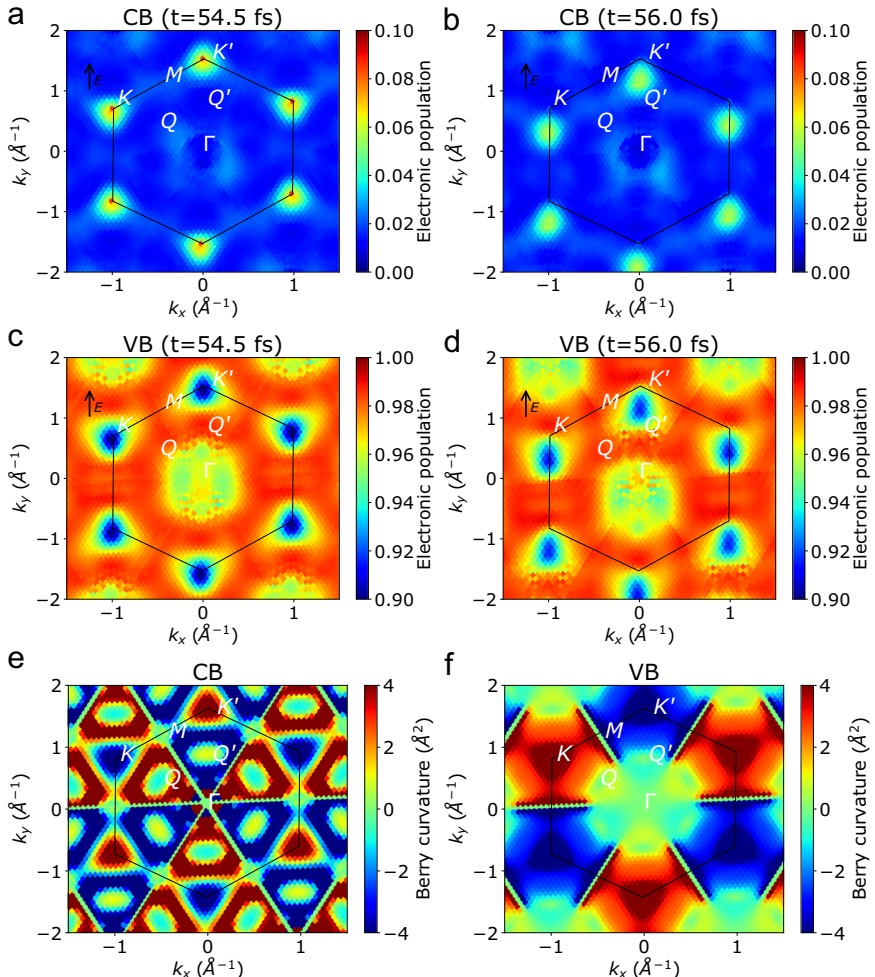

**Fig. 5 | Band occupation and Berry curvature. a** Electron occupation in **k** space of the bottom conduction band and **c** top valence band at the peak of the electric field computed using TD-aGW for a 30° field and a photon energy of 0.335 eV. **b** Electron occupation in **k** space of the bottom conduction band and **d** top valence band when the field reaches zero computed using TD-aGW. Black arrows indicate the direction of the electric field. **e** Berry curvature of monolayer $MoS_2$ for the bottom conduction band and **f** top valence band.

We find that we can only accurately reproduce experimentally observed monotonically decreasing parallel harmonics and nonmonotonic even harmonics if we include many-body exciton effects through the GW self energy. We interpret this difference between the parallel and perpendicular configurations as a consequence of the interplay of an excitonic enhancement of the oscillator strength for interband transitions and renormalization of the density of states in tandem with the intraband anomalous velocity arising from the Berry curvature. The restrictions on the carrier motion imposed by the Berry curvature may lead to enhanced electron-hole correlation effects, especially in the perpendicular configuration.

## Methods

Here, we develop an ab initio theoretical approach within the time-dependent adiabatic GW (TD-aGW)[44] formalism to understand many-body effects in HHG in the nonperturbative regime. In general, nonlinear optical phenomena in real materials can be understood based on the real-time propagation of the nonequilibrium interacting Green's function on the Keldysh contour[44,55–57]

$$\left[ i\frac{d}{dt} - H_0 + e\mathbf{E}(t) \cdot \mathbf{r} \right] G(t,t') = \delta(t,t') + \int_c \Sigma(t,\bar{t}) G(\bar{t},t') d\bar{t}. \quad (1)$$

Here, $H_0$ is the electronic mean-field Hamiltonian at equilibrium; $\mathbf{E}(t)$ is the external electric field at time $t$, which couples to the system in the length gauge through the position operator $\mathbf{r}$; $G$ is the contour-ordered two-time Green's function; and $\Sigma$ is the electron self-energy. There is an equivalent adjoint equation for the time-evolution over $t'$. In practice, the self-energy in conventional solids is well-approximated by the GW self-energy ($\Sigma^{GW} = iGW$, where $W$ is the screened Coulomb interaction[45]), and the evolution over $t$ and $t'$ can be decoupled by splitting the self-energy into an equilibrium contribution, $\Sigma^{GW}[G_0]$, and an instantaneous correction term, $\delta\Sigma^{GW}[G] = \Sigma^{GW}[G] - \Sigma^{GW}[G_0]$, where $G_0$ is the non-interacting Green's function[44,57]. We refer to this technique as the time-dependent adiabatic GW (TD-aGW), and the equation of motion can be rewritten in terms of the single-particle density matrix in the quasiparticle basis, $\rho_{nm,\mathbf{k}} \equiv \langle n\mathbf{k}|\rho|m\mathbf{k}\rangle$, as

$$ih\frac{\partial}{\partial t}\rho_{nm,\mathbf{k}}(t) = [H_0 - e\mathbf{E}\cdot\mathbf{r} + \Sigma^{GW} + \delta\Sigma(t),\rho]_{nm,\mathbf{k}} \quad (2)$$

where $n$ and $m$ are the band indices and $\mathbf{k}$ is a $\mathbf{k}$-point in the Brillouin zone. $\Sigma^{GW}$ is the self-energy at equilibrium computed within the GW approximation and the correction $\delta\Sigma$ is calculated within a Coulomb-hole static screened-exchange scheme (static-COHSEX)[45,57,58]. Additional details can be found in the Supplementary Information Note 1.

In this formalism, one challenge of obtaining the nonlinear response is the calculation of the intraband coupling terms, $[r^{intra}, \rho]_{nmk'}$. In previous work, this is accessed using the dynamical Berry phase[40]. Here, we rotate to a locally smooth gauge following our previous work in ref. 44, allowing us to treat interband and intraband terms on the same footing.

The time-dependent current is found by taking the trace of the time-dependent density matrix with the velocity operator **v**

$$\mathbf{J}(t) = \mathbf{Tr}(\rho(t)\mathbf{v}), \qquad (3)$$

and the corresponding high harmonic contributions to the current are computed by taking the Fourier transform of the time-dependent current[59–62],

$$HHG(\omega) = |\omega \int dt\, \mathbf{J}(t) e^{-i\omega t}|^2 \qquad (4)$$

We note that the Keldysh formalism with the full dynamics over two times has been previously applied to study the effects of defect and phonon scattering on HHG[59], and the static-COHSEX self energy within TD-aGW, which is suitable for describing exciton effects, has previously been applied to look at second and third harmonic generation in the perturbative regime, using a dynamic Berry phase approach[40–43].

## Data availability
The raw data used generate the figures during the current study have been deposited in the Materials Data Facility under accession code 10.18126/72ev-9181 and are also available from the corresponding authors by request.

## Code availability
Code used in the current study are available from the corresponding authors on request.

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

## Acknowledgements

This work was primarily supported by the National Science Foundation (NSF) Condensed Matter and Materials Theory (CMMT) program under Career Grant Number DMR-2337987 (D.Y.Q.). Initial development of the computational method was supported by the NSF CMMT program under Grant Number DMR-2114081 (D.Y.Q.). Development of the BerkeleyGW code was supported by Center for Computational Study of Excited-State Phenomena in Energy Materials (C2SEPEM) at the Lawrence Berkeley National Laboratory, funded by the U.S. Department of Energy, Office of Science, Basic Energy Sciences, Materials Sciences and Engineering Division, under Contract DE-AC02-05CH1231 (D.Y.Q.). Work at LSU was supported by the National Science Foundation, under Grant No. PHY-2110317. The calculations used resources of the National Energy Research Scientific Computing (NERSC), a DOE Office of Science User Facility operated under Contract DE-AC02-05CH11231; Anvil at Purdue University through allocation PHY230053 from the Advanced Cyberinfrastructure Coordination Ecosystem: Services & Support (ACCESS) program, which is supported by National Science Foundation grants #2138259, #2138286, #2138307, #2137603, and #2138296[63]; and the Texas Advanced Computing Center (TACC) at The University of Texas at Austin. We thank V. Korolev, M. Zuerch and D. Kartashov for helpful discussion. We thank H. Liu for providing raw experimental data used in Fig. 4.

## Author contributions

D.Y.Q conceived and supervised the project. V.C.L., Y.-H.C. and D.Y.Q. developed the theoretical methods and performed the calculations. V.C.L., L.Y, M.B.G., Y.-H.C, and D.Y.Q. analyzed the data. V.C.L. and D.Y.Q. wrote the paper and all authors contributed to the revision of the manuscript.

## Competing interests

The authors declare no competing interests.
