## [Peer Review File · Nature Communications]

REVIEWER COMMENTS

Reviewer #1 (Remarks to the Author):

The paper by Victor Chang Lee, Lun Yue, Mette B. Gaarde, Yang-hao Chan, and Diana Y. Qiu delves into the nonlinear optical response and High-Harmonic Generation (HHG) in Mono-layer MoS₂. The theoretical investigation focuses on the effects of Many-Body interaction using the time-dependent adiabatic GW (TD-aGW) approximation. The unperturbed electronic states are described using the GW approximation, and dynamic many-body effects are incorporated via an instantaneous correction to the self-energy. The study reveals that Many-Body interaction significantly alters the nonlinear response of MoS₂, particularly for anomalous harmonics, which are influenced by the substantial Berry curvature magnitude in this material. The prevailing approach in this field is grounded in the independent particle (IP) approximation. However, the IP framework, while successful in various aspects, lacks the reliability needed for materials where excitonic effects are prominent. The manuscript addresses an intriguing topic relevant to researchers investigating nonlinear optical effects in novel nano-structures, and it offers valuable insights into the effects of many-body interactions on the harmonic generation process. While the authors' approach is promising and applicable to other materials, there are certain aspects that require clarification and refinement:

1. Derivation of Dynamical Equation: The derivation of the main dynamical equation (Eq. 2) from Eq. (1) and the approximations made in this process need clarification. It would be beneficial to explicitly state whether a static retarded approximation is employed for the self-energy and address its validity, particularly in the context of screening timescales. In particular, the screening is built up on a timescale given by the electron-hole plasma oscillation period. The dynamic screening will set in when the plasma oscillation period is much smaller than the interaction time.

2. Comparison with Previous Work: The paper relies on a previous work (Phys. Rev. B 84, 245110, 2011), and the differences between the two should be clearly articulated. Specifically, is the primary distinction the use of GW+BSE instead of $G_{\{0\}}W_{\{0\}}+BSE$?

3. Calculation of $\delta\Sigma$: The details of calculating $\delta\Sigma$ within a Coulomb-hole static screening exchange scheme (static-COHSEX) should be made transparent. Providing comprehensive information is essential for understanding the approximations made.

4. Phenomenological Treatment of Relaxation Processes: The manuscript mentions the consideration of relaxation processes in a phenomenological manner. It is suggested to explore whether, at this

level of theory, these processes, particularly diagonal terms, could be incorporated more rigorously rather than phenomenologically, since self-energy contains some of this information.

5. Quasi-classical Explanation of Harmonic Behavior: I understand that in this level of theory it is difficult to outline simple physical picture, since dynamics is hidden behind complicated equations. The quasi-classical explanation for the different behavior of parallel and perpendicular harmonics raises some questions. The authors propose that the importance of electron-hole interactions diminishes in the parallel direction, while in the perpendicular direction, the Berry curvature acts as a pseudo-magnetic field, making electron-hole interaction crucial.

In the quasi-classical picture we have 3 amplitudes of the HHG signal:

$$\text{HHG} \sim \text{Elec.Hole_Creation}(k, t') * \text{Propagation}(k, t', t) * \text{Elec.Hole_annihilation}(k, t),$$

which have a transparent physical interpretation in analogy with the atomic three-step model: electron-hole creation at t' with the amplitude $\text{Elec.Hole_Creation}(k, t')$, then propagation in the BZ, which is defined by the classical action (including Berry curvature effects). Finally, the electron-hole pair annihilates at t with the amplitude $\text{Elec.Hole_annihilation}(k, t)$. In the propagation part electron-hole motion in parallel and perpendicular directions are interconnected and HHG signal is maximized when after the laser acceleration, the electron and hole collision/annihilation take place (irrespective that electrons and holes are driven in opposite directions in the beginning of interaction). Hence at the electron and hole collision/annihilation Coulomb effect will be crucial for both parallel and perpendicular HHG signal. I think that one should consider amplification in perpendicular direction due to $\text{Elec.Hole_annihilation}(k, t)$ amplitude, which contains Berry curvature.

6. Convergence of Results: The paper employs 8 valence bands and 6 conduction bands on a $36 \times 36 \times 1$ k grid. The authors should clarify whether this grid density is sufficient to capture the main physics, especially considering that many publications on the independent particle (IP) level often require a denser grid for convergence. Additionally, it would be beneficial to discuss the necessity of using 8 valence bands and 6 conduction bands, as the main dynamics appear to be captured by 2 valence bands (due to spin-orbit coupling) and 1 conduction band.

7. Minor remark: In the caption of FIG. 2. the polarization of the driving field and the direction of the measured HHG are described by the red arrows. I think one of them should be blue.

Addressing these points will enhance the clarity, rigor, and applicability of the presented work.

Dr. Garnik F. Mkrtchian

Leading Researcher,

Centre of Strong Fields Physics at Research Institute of Physics,

Yerevan State University,

Yerevan 0025, Armenia

e-mail: mkrtchian@ysu.am

Reviewer #2 (Remarks to the Author):

In their manuscript on many-body effects in HHG in MoS₂ the authors try to convince the reader that best agreement between experimental and theoretical spectra can be reached when going beyond the single particle picture. To calculate the bandstructure of the material this is certainly true (Fig. 1), when getting to directional high harmonics spectra (Fig. 2) I am not convinced. E.g., only DFT seems to be able to reproduce the symmetry of the 12th harmonic of the experimental data in the perpendicular configuration with a minimum around 30 deg where the other simulated results show the maximum of the distribution. It is, however, difficult to compare the intensities of the spectra as they are given on an undetermined scale (lin? log? a.u = atomic or arb. units? If the latter scaled to what? How does it compare to the scale of the experimental data?). Later, only relative differences between theoretical data are shown (Fig. 3) which does not help in assessing the validity of the underlying methods. The level of approximation that produces the highest intensity shifts from harmonic to harmonic with no clear trend visible - why should the reader trust any method more than the other? On the qualitative level of the current presentation not even the claimed (non-)monotonicity of harmonic intensities can be visually extracted from Figs. 3 and 4 (to my old eyes DFT performs equally "well" however that may be defined).

In summary, the authors have performed a formidable task to include many-body effects into the simulation of HHG from single-layer materials but have failed to convince me that this is indeed necessary to improve agreement with experimental data. I therefore do not see the required broad interest in the results and would recommend to transfer the material in a much extended version to a more specialized journal.

We thank the reviewers for their careful review and helpful comments, which have helped us greatly improve the manuscript. Below, we provide a summary of changes and a point-by-point response to reviewer comments. For clarity, reviewer comments are in bold and our response is unbolded. Changes to the manuscript are highlighted in red.

Summary of Changes

1. On Page 2, we have added additional theoretical details describing how the calculation of the intraband position operator on a smooth gauge allows for the calculation of nonlinear optical response.
2. In the caption of Fig. 2, we have clarified that our calculation is in atomic units, while the experimental data is in rescaled arbitrary units whose absolute intensity cannot be directly compared with our calculation.
3. In Fig. 3 and 4, we have replaced the figures with new figures showing the full HHG spectrum at specific angles. We have added a solid line connecting the peaks of odd or even harmonics as a guide to the eye, and we have added additional points from experimental data published in the literature.
4. In Fig. 4c-d, we have performed additional calculations to match the experimental conditions in the work of Yoshikawa et al in Ref. 18.
5. On page 4, we have added extensive discussion about the new figures.
6. In the SI, we have added additional data about k-point ad band convergence.
7. In the SI, we have added additional discussion of the static screening.

Reviewer #1

The paper by Victor Chang Lee, Lun Yue, Mette B. Gaarde, Yang-hao Chan, and Diana Y. QiuT delves into the nonlinear optical response and High-Harmonic Generation (HHG) in Mono-layer MoS₂. The theoretical investigation focuses on the effects of Many-Body interaction using the time-dependent adiabatic GW (TD-aGW) approximation. The unperturbed electronic states are described using the GW approximation, and dynamic many-body effects are incorporated via an instantaneous correction to the self-energy. The study reveals that Many-Body interaction significantly alters the nonlinear response of MoS₂, particularly for anomalous harmonics, which are influenced by the substantial Berry curvature magnitude in this material. The prevailing approach in this field is grounded in the independent particle (IP) approximation. However, the IP framework, while successful in various aspects, lacks the reliability needed for materials where excitonic effects are prominent. The manuscript addresses an intriguing topic relevant to researchers investigating nonlinear optical effects in novel nano-structures, and it offers valuable insights into the effects of many-body interactions on the harmonic generation process. While the authors' approach is promising and applicable to other materials, there are certain aspects that require clarification and refinement:

We thank the reviewer for their evaluation of the “valuable insights” provided by our work and their many thoughtful suggestions and questions, which have helped us improve the paper. In response to both reviewers’ questions we have made major changes to clarify the presentation of the data in Figs. 3 and 4. We have replaced histograms showing only the peaks in the harmonic yield with the full energy-dependent spectrum. Additionally, we have added comparison with experimental data from Ref. [Nat. Comm. 10, 3709 (2019)] and [Nature Phys. 13, 262–265 (2017)] and lines connecting odd or even peaks in the parallel or perpendicular spectra, respectively, as a guide to the eye. We have also added additional theoretical derivations and convergence data to the Supplemental Information (SI). Below, we address the reviewer’s comments point-by-point, highlighting corresponding revisions in the manuscript.

1. Derivation of Dynamical Equation: The derivation of the main dynamical equation (Eq. 2) from Eq. (1) and the approximations made in this process need clarification. It would be beneficial to explicitly state whether a static retarded approximation is employed for the self-energy and address its validity, particularly in the context of screening timescales. In particular, the screening is built up on a timescale given by the electron-hole plasma oscillation period. The dynamic screening will set in when the plasma oscillation period is much smaller than the interaction time.

We thank the reviewer for raising this interesting point. In the derivation of Eq. 2, the full frequency dependence is included within a plasmon pole model for the equilibrium part of the self energy, and a static approximation is used for the non-equilibrium self energy, $\delta\Sigma$. We justify this approximation based on our understanding of the dynamical screening in the Bethe Salpeter equation (BSE), which is dominated by frequencies close to the exciton binding energy [Strinati, PRL **20** 1519 (1982)]. In the perturbative regime, the plasma frequency is much higher than the typical exciton binding energy, so the screening is well-approximated as static. In the non-perturbative regime, as the reviewer points out, the picture could be more complicated, but overall, since we pump in the gap, the excited state population is small, and we estimate that the plasma frequency due to the excited-state population is ~ 0.02 meV, which is very far from the exciton binding energy of ~ 0.5 eV. This suggests that the static approximation is well-justified. Explicit inclusion of the dynamical screening would nonetheless be valuable to understand, but we believe it goes beyond the scope of the current work, which serves as a first demonstration of an *ab initio* nonequilibrium Green’s function approach to HHG.

2. Comparison with Previous Work: The paper relies on a previous work (Phys. Rev. B **84, 245110, 2011), and the differences between the two should be clearly articulated. Specifically, is the primary distinction the use of GW+BSE instead of $G_{\{0\}}W_{\{0\}}+BSE$?**

We would like to clarify that the equilibrium GW self energy that goes into Eq. 2 in our calculation is the one-shot G0W0 self energy, and some amount of self consistency comes in through the time-evolution. In this sense, our work follows the same basic methodology for

the time-evolution of the density matrix as that developed in the cited work by Attacalite and co-workers, and there is no functional distinction between GW+BSE and G0W0+BSE, only a different naming convention. However, the previous work is unable to directly access nonlinear optical properties due to a gauge uncertainty that prevents direct calculation of intraband matrix elements. One of the primary technical advances in our work is the development and application of a locally smooth gauge for the calculation of intraband transition matrix elements. In perturbative regime, one can calculate the nonlinear response using the dynamical Berry phase [Attacalite and Gruning, PRB, 2013], but to the best of our knowledge, there has been no successful attempt to connect this with Green's function theory or density matrix kinetic equations, allowing for its extension to the nonperturbative regime.

Changes to the manuscript: We have added the following sentences to the description of the theory on page 2 elaborating on this point:

“In this formalism, one challenge of obtaining the nonlinear response is the calculation of the intraband coupling terms. In previous work, this is accessed using the dynamical Berry phase [Attacalite and Gruning, PRB, 2013]. Here, we rotate to a locally smooth gauge following our previous work in Ref. [Chan, PNAS, 2021], allowing us to treat interband and intraband terms on the same footing”

3. Calculation of $\delta\Sigma$: The details of calculating $\delta\Sigma$ within a Coulomb-hole static screening exchange scheme (static-COHSEX) should be made transparent. Providing comprehensive information is essential for understanding the approximations made.

We have revised the discussion of the theory in the main manuscript to emphasize the static-COHSEX, and we add details about the calculation of $\delta\Sigma$ to the supplementary information (S1).

4. Phenomenological Treatment of Relaxation Processes: The manuscript mentions the consideration of relaxation processes in a phenomenological manner. It is suggested to explore whether, at this level of theory, these processes, particularly diagonal terms, could be incorporated more rigorously rather than phenomenologically, since self-energy contains some of this information.

We thank the reviewer for raising this important and interesting point about relaxation in the diagonal terms. We use an empirical parameter for the relaxation time, but in reality, as the reviewer notes, the relaxation time should be state-dependent, with higher energy states having a shorter relaxation time. However, this relaxation time is not trivial to incorporate from first principles because it involves both electron-electron and electron-phonon contributions. While the imaginary part of the self-energy contains the lifetime due to electron-electron scattering, it neglects electron-phonon scattering, which would take additional calculations of the electron-phonon self energy--a considerable computational challenge in itself. While the first principles study of relaxation is quite interesting, the literature suggests that HHG spectra are fairly insensitive to the diagonal relaxation time [A

Taghizadeh and T G Pedersen 2020 *2D Mater.* 7 015003]. Thus, we feel justified in taking an empirical value extracted from experiment, at least in this first calculation. We note that the use of an empirical relaxation time is also the standard approach for linear response GW+BSE calculations of optical absorption spectra.

5. Quasi-classical Explanation of Harmonic Behavior: I understand that in this level of theory it is difficult to outline simple physical picture, since dynamics is hidden behind complicated equations. The quasi-classical explanation for the different behavior of parallel and perpendicular harmonics raises some questions. The authors propose that the importance of electron-hole interactions diminishes in the parallel direction, while in the perpendicular direction, the Berry curvature acts as a pseudo-magnetic field, making electron-hole interaction crucial.

In the quasi-classical picture we have 3 amplitudes of the HHG signal:

HHG~Elec.Hole_Creation(k, t')*Propagation(k,t',t)*Elec.Hole_annihilation(k, t), which have a transparent physical interpretation in analogy with the atomic three-step model: electron-hole creation at t' with the amplitude Elec.Hole_Creation(k, t'), then propagation in the BZ, which is defined by the classical action (including Berry curvature effects). Finally, the electron-hole pair annihilates at t with the amplitude Elec.Hole_annihilation(k, t). In the propagation part electron-hole motion in parallel and perpendicular directions are interconnected and HHG signal is maximized when after the laser acceleration, the electron and hole collision/annihilation take place (irrespective that electrons and holes are driven in opposite directions in the beginning of interaction). Hence at the electron and hole collision/annihilation Coulomb effect will be crucial for both parallel and perpendicular HHG signal. I think that one should consider amplification in perpendicular direction due to Elec.Hole_annihilation(k, t) amplitude, which contains Berry curvature.

The reviewer raises an interesting point. We entirely agree that regardless of the direction, the electron and hole must be in close proximity when they annihilate, making the Coulomb interaction between them important for enhancing the magnitude of HHG in both the parallel and perpendicular directions. This is very much analogous to exciton enhancement of linear absorption, and indeed, it would not be enough to account for the difference in the parallel and perpendicular HHG.

We also agree with the reviewer that the Berry curvature is responsible for the generation of the perpendicular components of the HHG spectra observed as even order harmonics as explained in previous works (Nature volume 593, 385–390 (2021), Nature Physics volume 13, 262–265 (2017)), and in that sense, it contributes to enhancement of the even harmonics.

In our work, what we try to explain with the semiclassical picture is not necessarily enhancement, but why the interacting and noninteracting picture are qualitatively very different for perpendicular harmonics, but not parallel harmonics. Since the Berry curvature of valence and conduction bands have opposite signs, the electron and hole generated by the laser field have a real-space trajectory in the same direction for the perpendicular component,

so the effect of the electron-hole interaction builds up more for the perpendicular component (beyond just during annihilation).

This additional electron-hole interaction in the propagation steps that are included in our calculations allow us to accurately reproduce the nonmonotonic enhancement of *specific* harmonics seen in experiment. However, it does not lead to a universal enhancement of all harmonics.

6. Convergence of Results: The paper employs 8 valence bands and 6 conduction bands on a $36 \times 36 \times 1$ k grid. The authors should clarify whether this grid density is sufficient to capture the main physics, especially considering that many publications on the independent particle (IP) level often require a denser grid for convergence. Additionally, it would be beneficial to discuss the necessity of using 8 valence bands and 6 conduction bands, as the main dynamics appear to be captured by 2 valence bands (due to spin-orbit coupling) and 1 conduction band.

We added in the SI additional information related to the convergence for both bands and k-points. We agree with the reviewer that k-point convergence could be quite challenging. In the new data in the SI, we show that at the independent particle level, the HHG spectrum is converged with k-point sampling up to 4 eV. Thus, the convergence of higher harmonics will require a denser grid, but our convergence tests show that for the harmonics of interest, a $36 \times 36 \times 1$ k grid is sufficient.

We also added in the SI a discussion related to convergence with the number of bands. While the main physics is indeed described mainly by the top valence and bottom conduction band, the complex interplay of the higher conduction band and lower valence band can also have a significant contribution to the HHG spectra, specially at higher energy as shown in previous work Phys. Rev. Lett. **129**, 147401. Our numerical implementation in a Bloch basis may also contribute to the relatively slow convergence with respect to bands. Since the Bloch bands are not inherently smooth, we need to include enough bands to account for any possible band crossings across the Brillouin zone.

7. Minor remark: In the caption of FIG. 2. the polarization of the driving field and the direction of the measured HHG are described by the red arrows. I think one of them should be blue.

Thank you for pointing this out. We have fixed the caption.

Reviewer #2

In their manuscript on many-body effects in HHG in MoS₂ the authors try to convince the reader that best agreement between experimental and theoretical spectra can be reached when going beyond the single particle picture. To calculate the bandstructure of the material this is certainly true (Fig. 1), when getting to directional high harmonics

spectra (Fig. 2) I am not convinced. E.g., only DFT seems to be able to reproduce the symmetry of the 12th harmonic of the experimental data in the perpendicular configuration with a minimum around 30 deg where the other simulated results show the maximum of the distribution. It is, however, difficult to compare the intensities of the spectra as they are given on an undetermined scale (lin? log? a.u = atomic or arb. units? If the latter scaled to what? How does it compare to the scale of the experimental data?). Later, only relative differences between theoretical data are shown (Fig. 3) which does not help in assessing the validity of the underlying methods. The level of approximation that produces the highest intensity shifts from harmonic to harmonic with no clear trend visible - why should the reader trust any method more than the other? On the qualitative level of the current presentation not even the claimed (non-)monotonicity of harmonic intensities can be visually extracted from Figs. 3 and 4 (to my old eyes DFT performs equally "well" however that may be defined). In summary, the authors have performed a formidable task to include many-body effects into the simulation of HHG from single-layer materials but have failed to convince me that this is indeed necessary to improve agreement with experimental data. I therefore do not see the required broad interest in the results and would recommend to transfer the material in a much extended version to a more specialized journal.

We thank the reviewer for pointing out that the comparison between our theoretical results and previous experiment was difficult to interpret, which has helped us to greatly improve the paper. In response, we have made significant changes to the presentation of the data to make the comparison much more explicit, and we have performed extensive additional calculations that allow us to compare our results against a wider range of experimental data. We hope that the better agreement between our many-body TD-aGW results and experiment compared to DFT now comes across clearly to the reader. We summarize and explain our approach below.

- We now note in the captions that all of our calculated spectra are presented in atomic units (a.u.) not arbitrary units.
- In Fig.2, we compare angle-resolved spectra against the experiment from Yue et al. [Phys. Rev. Lett. **129**, 147401 (2022)]. This is a difficult direct comparison because the experimental data is presented in rescaled arbitrary units. To quote from that paper: "The yields are in arbitrary units and rescaled to the maximum value. In Fig. 1 of the main text the color scales have been adapted for each panel to optimise the contrast to compare the relevant angular features. Since the used spectrometer is not intensity calibrated a relative amplitude comparison of harmonics is not feasible." Thus, the experimental figure is mainly useful for showing how the intensity varies with angle across each harmonic and not relative intensity between harmonics.
- Next, we comment on the differences between theory and experiment in Fig. 2. While the 12th harmonic in the experiment and IP@DFT model present a similar double hump structure around 30 degrees, the fact that the symmetry for this particular even harmonic is completely different to that of the other even harmonics (8th, 10th and 14th) is a bit strange since a change in symmetry is related to the involvement of higher energy bands and should thus also be observable in the 14th harmonic. We

thus attribute the accidental agreement of the perpendicular 12th harmonic in DFT with experiment to be an outlier and hope to convince the referee of this fact with our new Figs. 3 and 4.

- The agreement between theory and experiment becomes much clearer if we look at the relative intensities of different harmonics. Thus, we have added additional experimental data, where the spectra **are** calibrated to allow for comparison of HHG intensity across harmonics to Fig. 4.
- We have replaced the intensities in Fig. 3 and Fig. 4 with the entire spectrum, added lines connecting the peaks of odd or even harmonics as a guide to the eye, and added experimental data from Ref. [Nat. Comm. 10, 3709 (2019)] and [Nature Phys. 13, 262–265 (2017)], which were able to obtain calibrated data capturing the relative differences between different harmonics. We have also performed additional calculations corresponding to [Nat. Comm. 10, 3709 (2019)] to allow us to compare against a wider range of experimental data.. We hope that with the new presentation, it is now clear that our TD-aGW method is able to capture changes in the relative intensities of different harmonics seen in experiment, while the independent particle pictures do not. This is the main result of our paper.

Finally, while we disagree with the reviewer that DFT is able to perform “equally well” in this case, we would like to emphasize that as in the discussion of linear response, DFT frequently gets the right result due to a cancellation of error. For instance, in the linear optical response, there is a fortuitous cancellation of error between the underestimation of the band gap and the neglect of the exciton binding energy. As a consequence, DFT has historically been very effective in aiding in the understanding of excited-state properties, even though it is fundamentally a theory of the electronic ground state. However, for the same reason, it can also lead to misleading interpretations of the underlying physical processes, such as in the discussion of resonant transitions where the oscillator strengths and dispersion associated with correlated many-body states might be very different from independent-particle states, even if the transition energies are similar. The TD-aGW approach we develop here is fundamentally a more physically accurate description of driven nonequilibrium systems. Hence, it has predictive power that DFT does not, and the development of this technique could have similar impact for the interpretation of HHG spectra as the Bethe-Salpeter equation (BSE) has had for linear optical spectra.

We hope that with these revisions the reviewer will reconsider their evaluation of the broad interest of our manuscript.

REVIEWERS' COMMENTS

Reviewer #1 (Remarks to the Author):

The authors have responded satisfactorily to all of my comments and have made revisions accordingly. They have addressed additional aspects such as k-point analysis, band convergence, and static screening, providing a more comprehensive understanding of their findings. Furthermore, they have conducted extensive additional calculations, enhancing the comparison of their results with a wider range of experimental data. As previously noted in my review, the manuscript explores an intriguing topic relevant to researchers investigating nonlinear optical effects in novel nanostructures. It offers valuable insights into the influence of many-body interactions on the harmonic generation process, marking a significant advancement in the field. Therefore, I strongly recommend the publication of this manuscript in Nature Communications.

Reviewer #1 (Remarks on code availability):

I don't have access to the code to draw any conclusion.

Reviewer #2 (Remarks to the Author):

For the revision of their manuscript, the authors have performed a large number of calculations to answer the criticism raised by the referees and have adapted the text accordingly. Together with the improved readability of the figures I now consider the manuscript ready for acceptance and publication as it is.